# Electroacupuncture at Neurogenic Spots in Referred Pain Areas Attenuates Hepatic Damages in Bile Duct-Ligated Rats

**DOI:** 10.3390/ijms22041974

**Published:** 2021-02-17

**Authors:** Yoo Jung Yi, Do Hee Kim, Suchan Chang, Yeonhee Ryu, Sang Chan Kim, Hee Young Kim

**Affiliations:** 1Department of Physiology, College of Korean Medicine, Daegu Haany University, Daegu 42158, Korea; dldbwjd3906@naver.com (Y.J.Y.); dh861206@hanmail.net (D.H.K.); c64111915@gmail.com (S.C.); 2Korean Medicine Fundamental Research Division, Korea Institute of Oriental Medicine, Daejeon 34054, Korea; yhryu@kiom.re.kr; 3Medical Research Center, College of Korean Medicine, Daegu Haany University, Gyeongsan 38610, Korea; sckim@dhu.ac.kr

**Keywords:** liver injury, neurogenic inflammation, substance P, calcitonin gene-related peptide, electroacupuncture

## Abstract

Visceral pain frequently produces referred pain at somatic sites due to the convergence of somatic and visceral afferents. In skin overlying the referred pain, neurogenic spots characterized by hyperalgesia, tenderness and neurogenic inflammation are found. We investigated whether neurogenic inflammatory spots function as acupoints in the rat model of bile duct ligation-induced liver injury. The majority of neurogenic spots were found in the dorsal trunk overlying the referred pain and matched with locations of acupoints. The spots, as well as acupoints, showed high electrical conductance and enhanced expression of the neuropeptides substance P (SP) and calcitonin gene-related peptide (CGRP). Electroacupuncture at neurogenic spots reduced serum hepatocellular enzyme activities and histological patterns of acute liver injury in bile duct ligation (BDL) rats. The results suggest that the neurogenic spots have therapeutic effects as acupoints on hepatic injury in bile-duct ligated rats.

## 1. Introduction

Acupuncture is one of the oldest medical procedures and has been practiced for at least 2500 years. Acupuncture stimulates specific skin areas called acupuncture points or acupoints [1]. Oriental medicine describes how acupoints are associated with internal organs. Acupoints reflect the status of internal organs, and the internal organ disorders can be treated by stimulating the acupoints [1,2]. In support of this, we and others have shown that acupoints become hypersensitive under abnormal visceral conditions, and that stimulation of the acupoints can relieve the symptoms of the associated internal organs [3,4].

While many studies have attempted to find out the anatomical structure of acupuncture points, our previous studies showed experimental evidence that acupoints associated with internal organs can be identified as cutaneous neurogenic inflammatory spots (neurogenic spots or Neuro-Sps), which are produced by activation of somatic afferents in abnormal conditions of internal organs [3] and can be visualized experimentally in the skin by systemic injection of Evans blue dye (EBD). By using rat models of immobilization-induced hypertension and colitis, our recent studies revealed that the neurogenic spots, as well as acupoints, show high mechanical sensitivity, high electrical conductance and C-fber-mediated sensations [5].

Our previous studies showed that in the rat models of hypertension and colitis, neurogenic spots matched with anatomical location of heart or colon-associated acupoints and generated the therapeutic effects on heart or colon when stimulated [3,6]. The goal of the present study was to extend our previous findings that neurogenic spots are associated with the internal organs and function as acupoints [3]. By using the rat model of bile duct ligation-induced liver injury, we investigated whether (1) neurogenic spots appear on liver-associated acupoints, (2) the neurogenic spots are found in referred pain area, (3) the neurogenic spots show high conductance and neuropeptides release of SP and CGRP and (4) the stimulation of neurogenic spots generates the therapeutic effects on liver injury.

## 2. Results

A preliminary study was performed to see whether BDL could induce neurogenic inflammation in skin. The rats were subjected to a sham operation (*n* = 6) or BDL (*n* = 6). The rats were given intravenous EBD on the 3rd (*n* = 2), 6th (*n* = 2) and 9th (*n* = 2) day after surgery and sacrificed. Cutaneous neurogenic inflammatory sites (neurogenic spots or Neuro-Sps) started to appear on the dorsal trunk about five minutes after EBD injection and the blue-dyed spots were observed clearly on the 3rd day rather than the 6th and 9th day after BDL surgery. Upon the postmortem examination, cystic dilation of bile duct stump, bile-stained tissues and ascites were observed in BDL group but not the sham-operated group (Sham, Figure 1A). Thus, the 3rd day after BDL surgery was chosen for the next experiment.

### 2.1. Neurogenic Spots in the Skin in BDL Rats

Rat were divided into Sham (*n* = 6) and BDL (*n* = 6) groups. The animals were given intravenous EBD on the 3rd day after BDL or sham surgery. There were considerable variations in the locations of neurogenic spots among rats, as reported in our and other studies [3,7]. However, neurogenic spots in BDL group were observed most commonly in the dorsal trunk. The BDL groups displayed significantly increased numbers of neurogenic spots, compared to the Sham group (Figure 1B,C). These neurogenic spots matched with the acupoints such as BL18, BL20, BL21, BL22, BL25, GV4, LR4 and ST44. The spots were also found in the nonacupoint areas (NA; Figure 1D).

### 2.2. Referred Hyperalgesia at the Dorsal Trunk of BDL Rats

To see whether BDL induces referred hyperalgesia, mechanical sensitivity was measured at the dorsal trunk in another set of animals consisting of BDL (*n* = 7) and Sham (*n* = 5) rats (Figure 2A). In the Sham group, the average 50% withdrawal threshold of dorsal trunk area to von Frey stimuli was the von Frey value of 5.22, equivalent to a bending force of 16.6 g. Three days after BDL, mechanical thresholds were markedly decreased to the von Frey values of 4.59 ± 0.13 (3.91 ± 0.01 g) at the dorsal trunk (Figure 2B).

### 2.3. Increased Electrical Conductance at Neurogenic Spots

The pressure exerted by the probe on the skin is considered to be as one of potential confounders affecting the reproducibility and reliability of data on the measurement of electrical skin conductance [8]. To hold the electrode to the skin at a constant pressure, a conductance probe that could measure electrical currents and touch pressure simultaneously was constructed (Figure 3A). Skin conductance was estimated as the peak value of electrical current recorded while pressing the electrode at a pressure of 300 g (Figure 3B,C). To explore whether neurogenic spots exhibit a high conductance, electrical currents at neurogenic spots (*n* = 18) and non-neurogenic spots (*n* = 15) 5 mm apart from the neurogenic spots on the dorsal trunk were compared in BDL rats (*n* = 5). When the conductance probe was applied over the neurogenic spots or non-neurogenic spots at a constant pressure of 300 g, the neurogenic spots showed a higher electrical conductance than that the non-neurogenic spots (*p* = 0.004; Figure 3D).

### 2.4. Enhanced Expression of SP and CGRP at Neurogenic Spots

To see the expression of SP and CGRP at neurogenic spots, we compared the expression of CGRP and SP in the skins of neurogenic spots (*n* = 6) and non-neurogenic spots (*n* = 6) from BDL rats (*n* = 6). Significantly increased SP (Figure 4A) and CGRP (Figure 4C) fluorescence was found in the dermis of neurogenic spots than in that of non-neurogenic spots (*t*-test; *p* < 0.05 in Figure 4B,D).

### 2.5. Alleviation of Hepatic Injury by Repetitive EA Treatments at Neurogenic Spots

To explore whether the neurogenic spots have therapeutic effects like acupoints, the effects of electroacupuncture (EA) at neurogenic spots on BDL-induced liver injury were assessed. Animals were divided into Sham (sham surgery, *n* = 6), control (BDL rats, *n* = 6), EA at Neuro-Sps (EA at neurogenic spots in BDL rats, *n* = 6) and EA at Non-Neuro-Sps (EA at nearby sites in BDL rats, *n* = 6). The BDL rats showed significantly elevated levels of ALT, AST and TB, compared to sham-operated rats (two-way repeated ANOVA: group factor F = 4.366, *p* = 0.063; time factor F = 14.828, *p* < 0.01; interaction F = 14.307, *p* < 0.01). In particular, the levels of ALT and AST showed a peak increase at two day after BDL surgery. For EA groups, electroacupuncture was applied to neurogenic spots or non-neurogenic spots for 15 min daily for four days. Electrical stimulation of neurogenic spots significantly prevented the elevation of ALT and AST following BDL, while such effects were not seen in the rats given EA at Non-Neuro-Sps (two-way repeated ANOVA: group factor F = 4.792, *p* = 0.041; time factor F = 16.854, *p* < 0.01; interaction F = 16.517, *p* < 0.01) (Figure 5A–D). On the other hand, EA did not affect TP levels in BDL rats. The effects of EA at neurogenic spots on BDL-induced liver injury were evaluated histologically. While normal group showed regular arrangement and nonleakage of nuclei in hepatic structures, BDL rats displayed leakage of nuclei and irregular arrangement and increased volume of hepatocytes. The histological changes in the BDL rats were alleviated by repetitive EA treatments at neurogenic spots, but not EA at Non-Neuro-Sps (Figure 5E).

## 3. Discussion

In the present study, the rat model of BDL displayed significantly increased numbers of neurogenic spots. Anatomical locations of these spots corresponded with liver-associated acupoints. BDL induced referred hyperalgesia in the dorsal trunk. Increased electrical conductance and the neuropeptides SP and CGRP were found in the neurogenic spots. Electrical stimulation of neurogenic spots reduced serum hepatocellular enzyme activities and histological patterns of acute liver injury in BDL rats. The results suggest that neurogenic spots in referred pain area have similar characteristics of traditional acupoints and their stimulation can attenuate the development of liver damage following BDL.

Our previous studies revealed that in the rat models of hypertension or colitis the skin over acupoints displayed neurogenic inflammation [3,6], which was attributed to viscerosomatic convergence at the spinal cord segments [9]. In the present study, most neurogenic spots in BDL rats were found in the dorsal trunk, which is innervated by the same spinal segments (T7–T13) that innervate the liver [10], and those spots corresponded to location of acupoints such as BL18, BL20, BL21, GV3, ST44 and LR4. The acupoints are prescribed most frequently for liver or gastrointestinal disorders [11,12]. Based on Oriental medicine, BDL-induced liver injury can be interpreted as liver qi depression pattern/syndrome [13]. These results suggest that the neurogenic spots are found in the dermatome of segmentally related liver, and match with the location of acupoints commonly used for treatment of hepatic or gastrointestinal disorders or liver qi depression syndrome. Visceral pain frequently produces referred pain at somatic sites due to the convergence of somatic and visceral afferents on the same neuron in the sensory pathway. In skin overlying the referred pain, local tissue reactions are accompanied by hyperalgesia and tenderness [7]. Liver pain is one of common clinical occurrences and commonly referred to distant regions such as mid-back, epigastrium, supraclavicle and shoulder. The referred liver pain is generated by inflammation and mechanical irritation of the inferior pleura and peritoneum. The referred pain is somatic and carried primarily by the lower intercostal and subcostal nerves [14,15]. Consistently, the present study showed that mechanical hypersensitivity was found in the dorsal trunk in BDL rats, indicating generation of referred liver pain in the dorsal trunk (mid-back). Furthermore, most neurogenic spots were found in the dorsal trunk. This finding further indicates that neurogenic spots are generated in the referred pain areas from the liver in BDL rats. On the other hand, the sham group also showed neurogenic spots. The sham group was subjected to the same abdominal surgery as the BDL group, except for bile duct ligation. The ventral side of the liver was lifted, the bile duct was separated from the flanking portal vein and hepatic artery using forceps, and the abdominal incision was closed. Although it is not clear why the sham group showed neurogenic spots, we assume that the abdominal manipulation during surgery would cause the irritation of abdominal organs and induce the neurogenic spots in the associated skin.

Previous studies have suggested that acupoints have distinct electrical properties, including a high conductance/capacitance and low impedance/resistance [16]. In the present study, high electrical conductance was found in the neurogenic spots in BDL rats, compared to surrounding tissue. Furthermore, increased expression of neuropeptides SP and CGRP was observed in neurogenic spots over acupoints in immunohistochemistry. SP increases microvascular permeability and edema formation by activating neurokinin receptors, while CGRP stimulates CGRP1 receptors to dilate arterioles [17]. Our recent study showed that in rat models of hypertension or colitis, neurogenic spots over acupoints reveal the neuropeptides release to cause neurogenic inflammation, plasma extravasation and accumulation of subskin water content, resulting in high electrical conductance and low impedance [5]. Thus, we suggest that neurogenic spots in BDL rats show high electrical conductance as acupoints, which is attributed to increased expression of SP and CGRP.

In the present study, electroacupuncture at neurogenic spots in referred pain suppressed the enhanced levels of ALT and AST in BDL rats. Similar to our findings, previous studies have shown that electroacupuncture can reduce inflammatory responses and increased ALT levels in the rat models of hepatic injuries [18,19,20]. This may indicate that neurogenic spots have similar therapeutic effects of acupoints on hepatic injury. Our previous studies have suggested the mechanisms underlying the effects of neurogenic spot stimulation on hypertension and colitis in rats. In the rat model of hypertension, elevated SP levels during neurogenic inflammation increase the responses of sensory afferents to the needling of acupoints and trigger acupuncture signaling to generate inhibitory effects of acupuncture on hypertension [3,6]. The present result, showing increased SP at neurogenic spots, may suggest that SP plays a critical role in evoking acupuncture effects on hepatic injury. In our previous studies, effects of neurogenic spot stimulation on hypertension were blocked by pretreatment of naloxone and replicated by the administration of morphine [3]. Acupuncture at the neurogenic spots alleviates colonic inflammation and diarrhea via the endogenous opioid system [21,22]. Taken together, these findings suggest that increased SP during neurogenic inflammation triggers peripheral acupuncture signals and may recruit the endogenous opioid system to reduce hepatic injury, which remains to be elucidated.

In conclusion, BDL induces neurogenic inflammatory spots in referred pain areas, and neurogenic spots have therapeutic effects like acupoints on BDL-induced hepatic injury.

## 4. Materials and Methods

### 4.1. Animals

Male Sprague-Dawley rats (weight 260–280 g, Daehan Animal, Seoul, Korea) were used. All rats had free access to food and water and were maintained on a 12 h light-dark cycle. All procedures were carried out in accordance with the National Institutes of Health Guide for the Care and Use of Laboratory Animals and approved by the Institutional Animal Care and Use Committee (IACUC) at Daegu Haany University (Identification code: DHU2019-076, Approved Date: 01 October 2019).

### 4.2. The Rat Model of Bile Duct Ligation-Induced Liver Injury

Bile duct ligation (BDL) was performed as described previously [23]. Briefly, the rats were pretreated with a subcutaneous injection of gentamycin (50 mg/kg) and anesthetized with pentobarbital sodium (50 mg/kg, intraperitoneal). After an abdominal midline incision, the ventral side of the liver was lifted so that the hepatic hilus became clearly visible. The bile duct was separated from flanking portal vein and hepatic artery using forceps, and a suture was placed around the bile duct and secured with a surgical knot. The abdominal incision was closed. Sham rats were given the same procedure without bile duct ligation.

### 4.3. Detection of Neurogenic Spots in the Skin by Evans Blue Dye Injection

Neurogenic spots were visualized by intravenously injecting Evans blue dye (EBD; 50 mg/kg, 50 mg/mL saline; Sigma-Aldrich, MO, USA) as described previously [3]. Three days after BDL or sham surgery, the rat was anesthetized with isoflurane and the distal portion of the tail was dipped into 40 °C warm water for at least 30 s. EBD was then injected via the tail vein with a catheter (26 gauge) and skin color changes were observed up to 2 h after injection. The blue-dyed areas on the skin were sketched using body charts, photographed and compared with a human acupoint chart based on the trans-positional method, which places acupoints on the surface of animal skin corresponding to the anatomic sites of human acupoints [24]. The numbers of blue-dyed spots on the skin surface were counted.

### 4.4. Measurement of Electrical Skin Conductance

Skin conductance was measured as performed in our laboratory [5]. Experiments were carried out under constant humidity (40~60%) and temperature (22 ± 2 °C). The animal hair was shaved prior to the placement of electrode. To simultaneously measure conductance and the applied pressure, a device was constructed by coupling a force transducer (FT-100, iWorx/CB Sciences Inc., Dover, NH, USA) with an electrical conductance probe (3.7 mm diameter, stainless). The rats were anesthetized with isoflurane. While the positive electrode was attached to the tail surface, the device (negative electrode) was placed on the skin over neurogenic spots or non-neurogenic spots 5 mm apart from the neurogenic spots and was pressed at a force of 300 g. Signals from the conductance (current) probe and the force transducer were fed to an ETH-200 Bridge Amplifier (CB Scientific Inc., Pinellas Park, Fl,, USA) and a GSR AMP device (Model FE116, ADInstruments, Colorado Springs, CO, USA), respectively, and digitized through a PowerLab 4/30 acquisition system (ADInstruments). Conductance was estimated as the maximum current value (μA) when a pressure of 300 g on the electrode was applied.

### 4.5. Referred Hyperalgesia

Mechanical sensitivity on the dorsal skin was determined by measuring the withdrawal responses to the application of von Frey filaments to the skin of the dorsal trunk. Briefly, each animal was placed in a plastic chamber (8.0 × 9.0 × 24 cm) on top of a mesh screen platform. Three days after BDL or sham operation, mechanical thresholds were assessed by the up-down method [25] using a set of von Frey monofilaments (VF). A withdrawal response was defined as trunk shakes, scratching or immediate licking of the site of application of hair. The data were expressed using a linear scale in von Frey values.

### 4.6. Immunohistochemistry for SP or CGRP in the Skin

Three days after BDL surgery, rats (*n* = 6) were given EBD injection and neurogenic spots (*n* = 6) and non-neurogenic spots 5 mm apart from the neurogenic spots (Non-Neuro-Sp, *n* = 6) were taken. The skin samples were paraffin-embedded, sectioned at 5 μm thickness and incubated with either anti-SP mouse monoclonal antibodies (1:500; GeneTex International Corporation, Irvine, CA, USA) or anti-CGRP mouse antibodies (1:500; Chemicon International Inc., CA, USA), followed by incubation with secondary antibody (1:500, Alexa Fluor 488-conjugated donkey antimouse IgG antibody, Thermo Scientific, Waltham, MA, USA). The sections were mounted on gelatin-coated slides and cover-slipped. Skin images were taken from three sections from each skin with an epifluorescence microscope (Olympus BX51, Tokyo, Japan) and quantified by using ImageJ software (National Institute of Mental Health, Bethesda, MD, USA). The mean intensity of green fluorescence was measured.

### 4.7. Electroacupuncture at Neurogenic Spots

Acupuncture stimulation was performed as described previously [3]. In brief, under light isoflurane anesthesia (1.5% isoflurane in 100% oxygen), acupuncture needles (diameter 0.10 mm, length 10 mm; Dongbang Acupuncture Inc., Qing Dao, China) were inserted at a depth of 10 mm into 6 neurogenic spots or 6 non-neurogenic spots (Non-Neuro-Sp) 5 mm apart from the neurogenic spots in the dorsal trunk and stimulated at a condition of 3 Hz and 0.5 mA for 15 min daily for four days after BDL by using an electrostimulator (Model H-306, Han-il Co, Seoul, Korea).

### 4.8. Blood Biochemical Analysis

Prior to electroacupuncture (EA) treatment, the rats were anesthetized with isoflurane (1.5% isoflurane in 100% Oxygen) and the blood samples (1–1.5 mL) were taken through jugular vein and kept in EDTA-coated bottles. The samples were centrifuged at 3500 rpm for 15 min to obtain plasma. Alanine transaminase (ALT), aspartate transaminase (AST) and total bilirubin (TBIL) were measured using the VetTest Chemistry Analyzer 8008 (IDEXX Laboratories Inc., Westbrook, ME, USA).

### 4.9. Histology

At the termination of EA experiments (4th day after BDL), all rats were sacrificed for histological examination. Animals were perfused with phosphate-buffered saline (PBS) and then with 4% paraformaldehyde. Livers were removed, postfixed in 4% paraformaldehyde, embedded in paraffin-blocks, sectioned into 4 μm-thick, stained with Hematoxylin & Eosin (H&E) and examined under the microscope (Olympus BX51, Tokyo, Japan).

### 4.10. Statistical Analysis

All data are presented as mean ± standard error of the mean (SEM) and analyzed by two-way repeated measures analysis of variances (ANOVA) followed by Tukey’s post hoc test or unpaired *t*-tests. *p*-values below 0.05 were considered statistically significant.

## Figures and Tables

**Figure 1 ijms-22-01974-f001:**
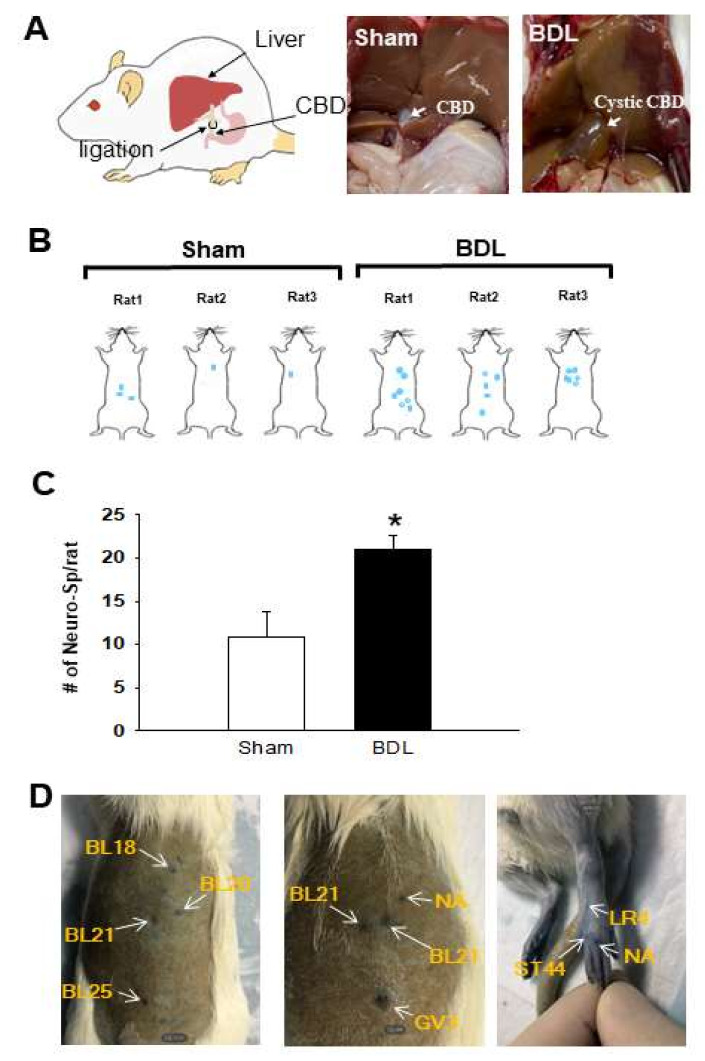
Neurogenic inflammation in the skin of BDL rats. (**A**). Schematic for bile duct ligation (BDL, left) and representative appearances of liver three days after sham operation (middle) and BDL (right). (**B**). Representative distribution of neurogenic spots visualized by EBD in Sham or BDL group. (**C**). Mean numbers of neurogenic spots per rat in sham or BDL rats. * *p* < 0.05 vs. Sham. (**D**). Representative pictures of neurogenic spots corresponding to acupoints. *n* = 6/group. EBD, Evans blue dye.

**Figure 2 ijms-22-01974-f002:**
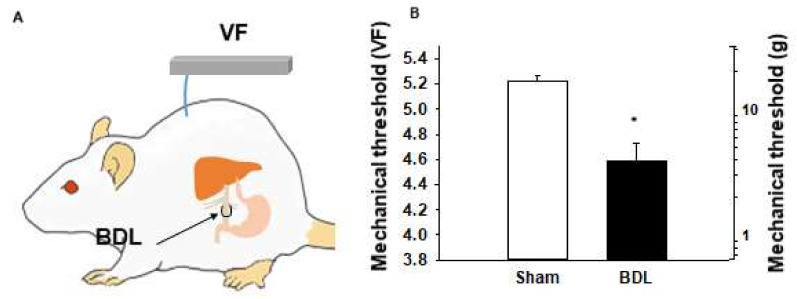
Increased mechanical sensitivity at the dorsal trunk in BDL rats. (**A**). Mechanical threshold was measured at the dorsal trunk by von Frey filaments (VF). (**B**). Comparison of mechanical sensitivity at the dorsal trunk in BDL (*n* = 7) or sham (*n* = 5) operated rats. * *p* < 0.05 vs. Sham. BDL, bile duct ligation.

**Figure 3 ijms-22-01974-f003:**
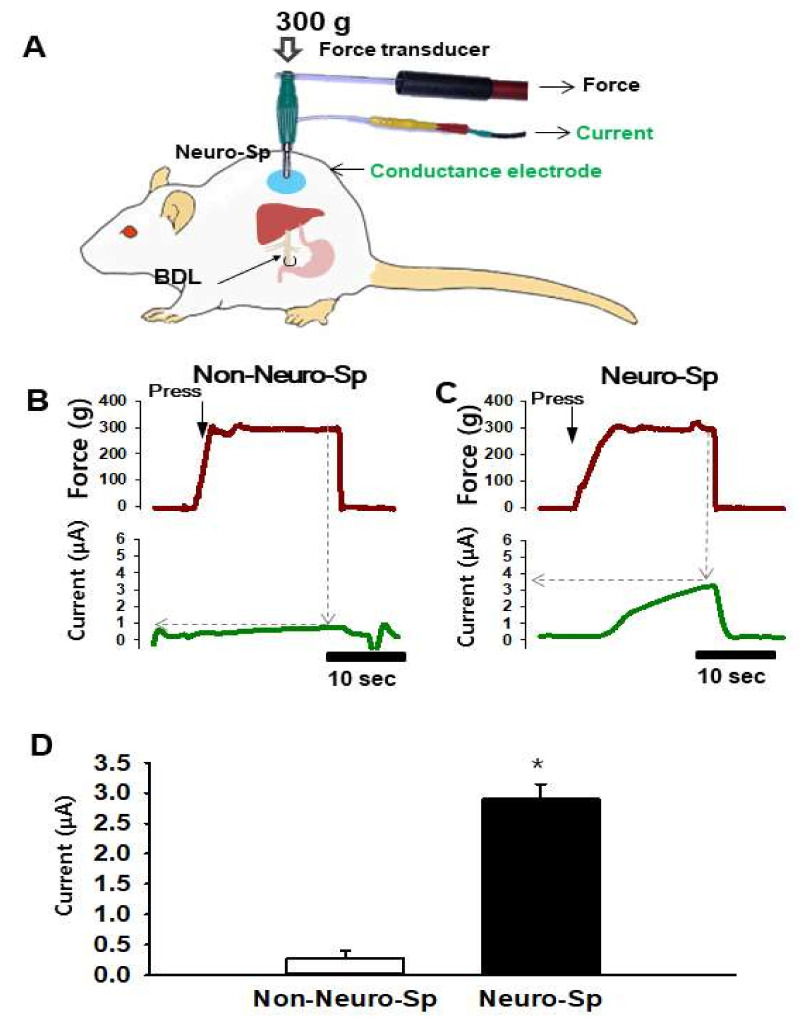
Increased electrical conductance at neurogenic spots in BDL rats. (**A**). Schematic representation for measurement of electrical conductance by a constructed electrode that enables the simultaneous measurement of conductance and applied pressure. (**B**,**C**). A representative trace of applied force (upper panels) and electrical currents (lower panels). Conductance was estimated as the maximum current value (μA) reached when a pressure of 300 g on the electrode was held constant. (**D**). Increased conductance at neurogenic spots (Neuro-Sp) in BDL rats, compared to non-neurogenic spots (Non-Neuro-Sp, 5 mm apart from neurogenic spots). * *p* < 0.05 vs. Non-Neuro-Sp. BDL, bile duct ligation.

**Figure 4 ijms-22-01974-f004:**
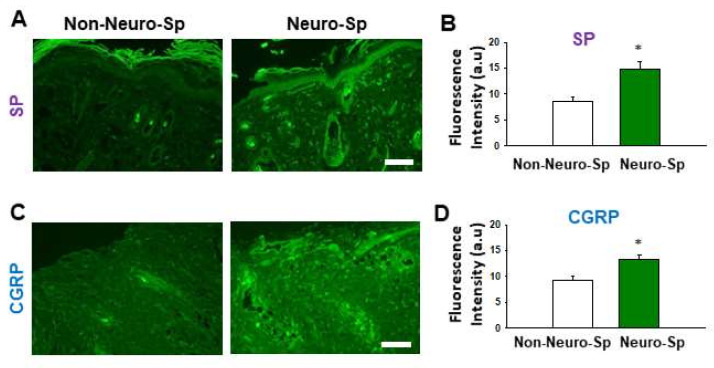
Increased expression of SP and CGRP at neurogenic spots. (**A**,**B**). SP expression at neurogenic spots (Neuro-Sp) and non-neurogenic spots (Non-Neuro-Sp) in BDL rats. (**C**,**D**). CGRP expression at neurogenic spots and non-neurogenic spots in BDL rats. * *p* < 0.05 vs. Non-Neuro-Sp. Bar = 100 μm. a.u = arbitrary unit. SP, substance P; CGRP, calcitonin gene-related peptide.

**Figure 5 ijms-22-01974-f005:**
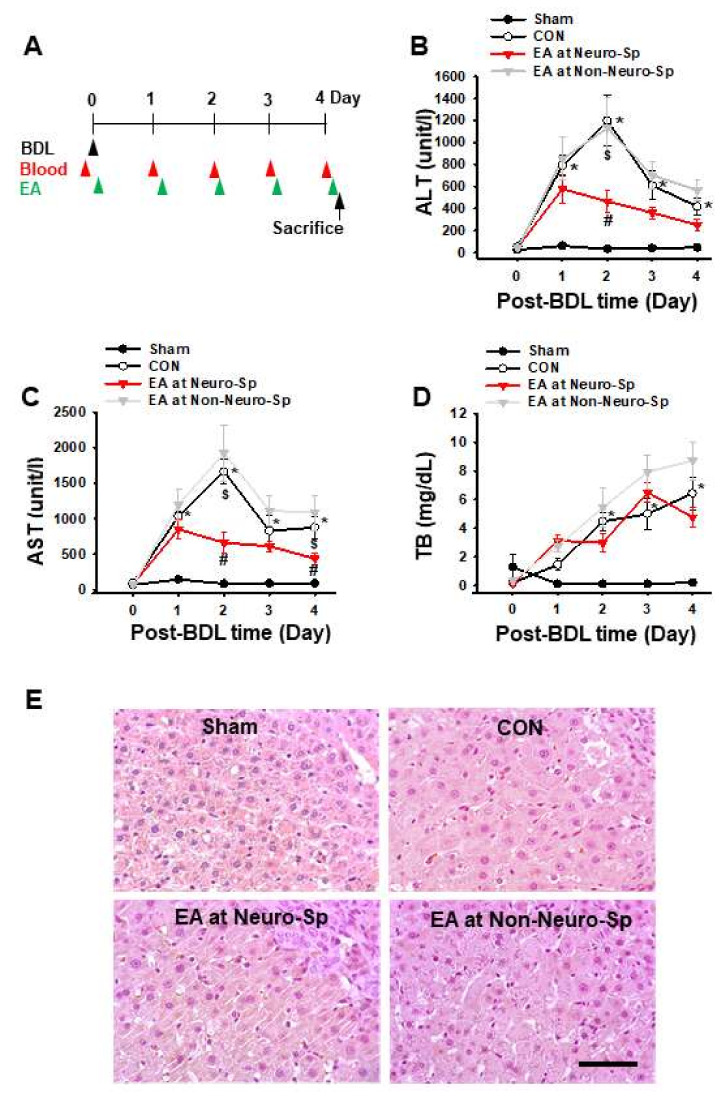
Effect of electroacupuncture at neurogenic spots on BDL-induced hepatic injuries. (**A**). Experimental schedule. BDL, bile duct ligation; Blood, blood sample from jugular vein; EA, electroacupuncture. (**B**–**D**). Effect of electroacupuncture at neurogenic spots on the levels of ALT (alanine transaminase, (**B**)), AST (aspartate transaminase, (**C**)) and TB (total bilirubin, (**D**)) in BDL rats. * *p* < 0.05, Sham vs. CON; ^#^
*p* < 0.05, CON vs. EA at Neuro-Sps, ^$^
*p* < 0.05, EA at Neuro-Sps vs. EA at Non-Neuro-Sp. *n* = 6/group. (**E**). Histological examination of EA effects at neurogenic spots on BDL-induced liver injury. Hematoxylin and Eosin staining of liver tissue. Bar = 50 μm. Sham (sham surgery), CON (BDL rats), EA at Neuro-Sps (EA at neurogenic spots in BDL rats) and EA at Non-Neuro-Sp (EA at nearby sites 5 mm apart from neurogenic spots in BDL rats).

## Data Availability

The data presented in this study are available within the article.

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
