# Peer review of "Electroacupuncture at Neurogenic Spots in Referred Pain Areas Attenuates Hepatic Damages in Bile Duct-Ligated Rats"

_ijms, 2021, doi:10.3390/ijms22041974_

Round 1
Reviewer 1 Report
This study showed a very interesting result of experimentally verifying the hypothesis of traditional acupuncture points, which are the main treatment sites for acupuncture, in bile duct-ligated rats. The following few modifications will help better understanding.
- P3 Electroacupuncture at neurogenic spots
I would like to know more about rationale for electroacupuncture (EA) treatment. Is "15 min per day" a typical dose for EA in animal experiments? Although it reflects the actual clinical trials in humans, it seems to be somewhat overstimulated to animals.
- Please check throughout the manuscript that the full name for the abbreviation is properly described. In particular, abstracts and figure legends must be properly abbreviated so that they can function independently.
- Was the sample size for each group in Figure 1B and 1C “n = 6/group”? Why were only 3 animals (per group) marked in Figure 1B?
- Was there a correlation between the locations of neurogenic spots in each individual in the BDL group? I would like to know more information about individual differences.
- How should the neurogenic spots observed in the sham group be interpreted? Is it difficult to say that the neurogenic spots of the sham group were related to acupuncture points? If so, what is the rationale for it? The mechanical threshold in Figure 2 was measured for the dorsal trunk area, not the neurogenic spots of the sham and BDL group, respectively, so it seems difficult to answer this question.
- Was the mechanical threshold measurement (Figure B) performed independently from the experiment in Figure 1 that observed the neurogenic spots? I was wondering why the sample size was different. In this case, can the spots between two independent experiments be considered to be identical?
- If there is a subtitle in the "results" like "methods", it will be helpful to understand.
- In the second paragraph of the P9 (Discussion), “(Kim DH, 2017, Fan Yu 2020)” should be converted to appropriate reference numbers. In particular, the bibliography on “Fan Yu 2020” was omitted from the reference list.
Author Response
Reviewer 1
Thank you for constructive comments. According to reviewer’s comments, our manuscript has been revised and the modification was indicated by using red font.
Point 1. “...Is "15 min per day" a typical dose for EA in animal experiments?...”
Response: Electroacupuncture of 15-30 min per session are generally used in rats. Many previous studies have also adopted a dose of “15-min” for electroacupuncture in rats (Chen et al., 2016; Oliveira and Prado, 2000; Su et al., 2019; Wu et al., 2013), as performed in our study.
Point 2. “Please check throughout the manuscript that the full name for the abbreviation is properly described. In particular, abstracts and figure legends must be properly abbreviated so that they can function independently.”
Response: As reviewer comments, the abbreviations are clarified, especially in abstract and figure legends.
Point 3. “Was the sample size for each group in Figure 1B and 1C “n = 6/group”? Why were only 3 animals (per group) marked in Figure 1B?”
Response: As described in Page 5, last line 2, the sample sizes are n=6/group. Figure 1B shows representative distribution of neurogenic spots in 3 of 6 rats. To avoid the confusion, the term “representative” is inserted into the legend of figure 1B.
Point 4. “Was there a correlation between the locations of neurogenic spots in each individual in the BDL group? I would like to know more information about individual differences.”
Response: As reported in our and other studies (Kim et al., 2017; Wesselmann and Lai, 1997), there are considerable variations in the locations of neurogenic spots among the rats. However, the neurogenic spots in BDL group were observed most commonly in the dorsal trunk. It is described in Page 5, lines 3-6 with addition of the supporting references #3 and #10.
Point 5. “How should the neurogenic spots observed in the sham group be interpreted? Is it difficult to say that the neurogenic spots of the sham group were related to acupuncture points? If so, what is the rationale for it? The mechanical threshold in Figure 2 was measured for the dorsal trunk area, not the neurogenic spots of the sham and BDL group, respectively, so it seems difficult to answer this question.”
Response: As reviewer pointed it out, sham group also showed neurogenic spots. Sham group was subjected to the same abdominal surgery as BDL group, except bile duct ligation. The ventral side of the liver was lifted, bile duct was separated from flanking portal vein and hepatic artery using forceps and the abdominal incision was closed. Although it is not clear why sham group showed neurogenic spots at both acupoints and non-acupoints, we assume that the abdominal manipulation during surgery would cause the irritation of abdominal organs and induce the neurogenic spots in the associated skin. It is described into Page 9, lines 30-37.
Point 6. “Was the mechanical threshold measurement (Figure B) performed independently from the experiment in Figure 1 that observed the neurogenic spots? I was wondering why the sample size was different. In this case, can the spots between two independent experiments be considered to be identical?”
Response: Yes, the mechanical threshold measurement was performed independently from the experiment of Figure 1. To avoid the confusion, the description of “another set of animals” is added into Page 5, line 13.
Point 7. “If there is a subtitle in the "results" like "methods", it will be helpful to understand.”
Response: As reviewer comments, the subtitles are added in the result section.
Point 8. “In the second paragraph of the P9 (Discussion), “(Kim DH, 2017, Fan Yu 2020)” should be converted to appropriate reference numbers. In particular, the bibliography on “Fan Yu 2020” was omitted from the reference list.”
Response: Thanks for pointing it out. It is corrected.
Additional references
Chen, L., Sun, H.X., Xia, Y.B., Sui, L.C., Zhou, J., Huang, X., Zhou, J.W., Shao, Y.D., Shen, T., Sun, Q., Liang, Y.J., Yao, B., 2016. Electroacupuncture decreases the progression of ovarian hyperstimulation syndrome in a rat model. Reprod Biomed Online 32, 538-544.
Kim, D.H., Ryu, Y., Hahm, D.H., Sohn, B.Y., Shim, I., Kwon, O.S., Chang, S., Gwak, Y.S., Kim, M.S., Kim, J.H., Lee, B.H., Jang, E.Y., Zhao, R., Chung, J.M., Yang, C.H., Kim, H.Y., 2017. Acupuncture points can be identified as cutaneous neurogenic inflammatory spots. Sci Rep 7, 15214.
Oliveira, R., Prado, W.A., 2000. Anti-hyperalgesic effect of electroacupuncture in a model of post-incisional pain in rats. Braz J Med Biol Res 33, 957-960.
Su, C., Chen, Y., Chen, Y., Zhou, Y., Li, L., Lu, Q., Liu, H., Luo, X., Zhu, J., 2019. Effect of electroacupuncture at the ST36 and GB39 acupoints on apoptosis by regulating the p53 signaling pathway in adjuvant arthritis rats. Mol Med Rep 20, 4101-4110.
Wesselmann, U., Lai, J., 1997. Mechanisms of referred visceral pain: uterine inflammation in the adult virgin rat results in neurogenic plasma extravasation in the skin. Pain 73, 309-317.
Wu, M.T., Shaw, L.H., Wu, Y.T., Tsai, T.H., 2013. Interaction of acupuncture and electroacupuncture on the pharmacokinetics of aspirin and the effect of brain blood flow in rats. Evid Based Complement Alternat Med 2013, 670858.
Reviewer 2 Report
Just a slight objection at the second sentence of the introduction "Oriental medicine describes how each acupoint is linked to a specific internal organ". The link is specific to organs dysfunctional status rather than straight to the organ itself: different syndromes require the stimulation of different acupoints. A little addjustment to the sentence will be appreciated by acupunture practitioners. Worth perhaps adding some comment about the Traditional Chinese Medicine interpretation of the dysfunction produced by the "bile duct ligation-induced liver injury" to better understand the connection to the acupoints mentioned in the conclusion
Author Response
Point 1. “Just a slight objection at the second sentence of the introduction "Oriental medicine describes how each acupoint is linked to a specific internal organ". The link is specific to organs dysfunctional status rather than straight to the organ itself: different syndromes require the stimulation of different acupoints. A little addjustment to the sentence will be appreciated by acupunture practitioners. Worth perhaps adding some comment about the Traditional Chinese Medicine interpretation of the dysfunction produced by the "bile duct ligation-induced liver injury" to better understand the connection to the acupoints mentioned in the conclusion”
Response: As reviewer comments, the modification has been made by using red-font in revised manuscript as follows;
(1) The sentence "Oriental medicine describes how each acupoint is linked to a specific internal organ" is changed into “Oriental medicine describes how acupoints are associated with internal organs; acupoints reflect the pathophysiological status of internal organs, and the internal organ disorders can be treated by stimulating the acupoints” in lines 3-5 in first paragraph of Introduction.
(2) Based on Traditional Chinese medicine, bile duct ligation-induced liver injury can be interpreted as “liver qi depression pattern/syndrome”. Thus this point is incorporated into discussion (Page 9, lines 14-18) with addition of a reference.
